# Tracing Superior Late-Leafing Genotypes of Persian Walnut for Managing Late-Spring Frost in Walnut Orchards

Mehdi Fallah [1], Mousa Rasouli [2], Darab Hassani [3], Shaneka S. Lawson [4], Saadat Sarikhani [1,*] and Kourosh Vahdati [1,*]

[1]  Department of Horticulture, College of Aburaihan, University of Tehran, Tehran 1417614335, PC, Iran
[2]  Department of Horticulture and Landscape Engineering, Faculty of Agriculture, Malayer University, Malayer 6571995863, PC, Iran
[3]  Temperate Fruits Research Center, Horticultural Science Research Institute (HSRI), Agricultural Research, Education and Extension Organization (AREEO), Karaj 3177777411, PC, Iran
[4]  USDA Forest Service, Hardwood Tree Improvement and Regeneration Center (HTIRC), Department of Forestry and Natural Resources, Purdue University, West Lafayette, IN 47907, USA
*   Correspondence: saadat.sarikhani@ut.ac.ir (S.S.); kvahdati@ut.ac.ir (K.V.)

**Abstract:** Evaluating genetic diversity in walnut (*Juglans regia* L.) populations is a rapid approach used by walnut breeding programs to distinguish superior genotypes. The present study was conducted on the walnut population of Hamedan Province, one of the richest and most genetically diverse regions in Iran, during 2018–2019. After the initial screening, 47 genotypes were selected for further evaluation of pomological and phenological traits based on International Plant Genetic Resources Institute (IPGRI) descriptors. Nut and kernel weights among the selected genotypes ranged from 7.15 to 21.05 g and 3.0 to 10.8 g, respectively. Principal component analysis (PCA) categorized the genotypes into three distinct groups. Whereas the cluster analysis (CA) revealed the similarities and dissimilarities among the genotypes by identifying four major clusters. Spearman correlation analysis showed a positive correlation ($p < 0.01$) between nut weight (NWT), nut size, and kernel weight (KW), while a negative correlation ($p < 0.01$) between shell thickness (STH) and packing tissue thickness (PTT) with kernel percentage (KP) was observed. Lastly, 10 of 47 genotypes (TAL8, TAL9, TAL10, TAL14, TAL19, TAL22, TB2, TB4, TB6, and RDGH5) were considered superior. Superior genotypes were late-leafing (25–40 days after the standard) and displayed a lateral bearing (LB) habit with heavy nuts (12.52–16.82 g) and kernels (6.53–8.15 g), thin shells (1.06–1.25 mm), and lightly colored kernels.

**Keywords:** walnut diversity; germplasm evaluation; late-leafing; pomological characteristics; superior genotypes





## 1. Introduction

Genetic diversity, which supports and complements species and ecological diversity, is the cornerstone of the three levels of biodiversity [1]. Genetic diversity protects species and ecosystems from sudden changes and allows them to adapt to changing environments, climates (consider the late-spring frost), and other challenges [1]. The large genetic diversity of walnuts in Iran might be related to the plant's climate variability [2].

Late-spring frost is a primary factor limiting walnut production in Iran and other temperate zones. Growers use several active and passive strategies to alleviate the negative effects of frost damage [3]. A strategy to limit damage severity resulting from this phenomenon is the use of late-leafing cultivars [4]. Therefore, the most efficient way to deal with late-spring frost is to consider genetic potential, which is a reliable and sustainable method [5–7].

Genetic diversity exploitation and germplasm screening are rapid breeding strategies utilized to address fruit-tree production challenges [8]. Genetic diversity among walnuts in Iran is both substantial and expected since the country serves as one of the species'

origins [9,10]. Thus, Iran is regarded as a rich natural source of walnut germplasm for the development of superior genotypes [10].

Spring-frost damage to fruit trees is determined by the budbreak date, that is, how soon after the first spring warming they bud break [11]. Fruit trees lose their frost resistance after bud break in response to rising temperatures in the spring. If the temperature decreases below or near freezing again, frost damage might happen [12]. Therefore, late-leafing and late-flowering dates reduce tree damage from the late-spring frost. The majority of Iran's north and northwest walnut orchards were damaged by the late-spring frost in 2018 [13]. To overcome this problem, breeding for late-leafing is an efficient strategy to mitigate the severe damage caused by the late-spring frost [4,14]. The natural abundance of rich walnut germplasm in Iran allows breeding programs to identify and develop late-leafing and superior commercial varieties [4,14]. It is highly likely late-leafing genotypes that are also late-spring frost-resistant exist within this vast germplasm. Hence, the evaluation of germplasm in the walnut breeding programs in Iran is very important.

The centralized walnut breeding program in Iran was started at the Institute of Horticultural Sciences Research Institute (HSRI) based on germplasm evaluation in 1983 and led to the development of Iran's first two walnut cultivars 'Jamal' and 'Damavand' (a pollinizer) [15]. Only 39 promising late-leafing and lateral bearing (LB) walnut genotypes were selected from more than 41,000 ha of walnut plantations in different regions of Iran during the second phase of the breeding program [16]. Phenological and pomological traits for the selected genotypes were studied until 2018. Then, in 2019, four new cultivars named 'Persia', 'Caspian', 'Alvand', and 'Chaldoran' were released [17]. In parallel with this work, other researchers evaluated seed-originated populations across the different regions of Iran [6,9,17–20]. Ebrahimi et al. [9] evaluated a seedling walnut population and introduced 14 genotypes out of 61 late-leafing trees selected. Another breeding trait study led to the introduction of the two late-leafing genotypes FaBaAg1 and FaBaAv2 [20]. In Turkey, a hybridization program between the 'Chandler' and 'Maras18' cultivars sought to develop late-leafing cultivars with superior nut traits and early harvest dates. Phenological and pomological evaluations resulted in the introduction of the 'Helete Gunesi' cultivar with a leafing date (LD) of 22 April, harvest date of 17 September, nut weight (NWT) of 13.41 g, and a kernel percentage (KP) of 53.39% [21].

The province of Hamedan in western Iran is the largest producer of walnuts. Like other provinces in the country, walnut trees are located in traditional open-pollination orchards. Therefore, considerable genetic diversity within breeding traits, especially LD, can be found in this population. A primary anthropogenic issue in this region, as in many worldwide, is late-spring frost. Our primary objective in this study is to identify a reliable method for the identification of superior and late-leafing walnut genotypes by exploiting the genetic diversity within the region. With these data, we hope to identify and develop populations resistant to or unaffected by late-spring frost.

## 2. Materials and Methods

### 2.1. Plant Materials

This research was conducted in the Hamedan province of Iran, a leading center of walnut genetic diversity, in 2018 and 2019. A total of 180 seed-originated walnut trees with 30–50-year-old genotypes located in traditional orchards were selected and pre-evaluated. The studied genotypes were selected from the main walnut-producing areas of Hamedan, including Tuyserkan, Razan, and Korzan based on phenotypic traits (Figure 1). The geographical location and climatic conditions for these areas were reported (Table 1 and Figure S1a–c) (https://www.worldweatheronline.com, http://www.sinamet.ir, accessed on 11 September 2021). The early selection looked for a certain height, form, location from another tree, and locales with late-spring frost. A total of 47 genotypes were chosen for further evaluation of phenotypic and pomological qualities. These genotypes were named based on region (Table 2).

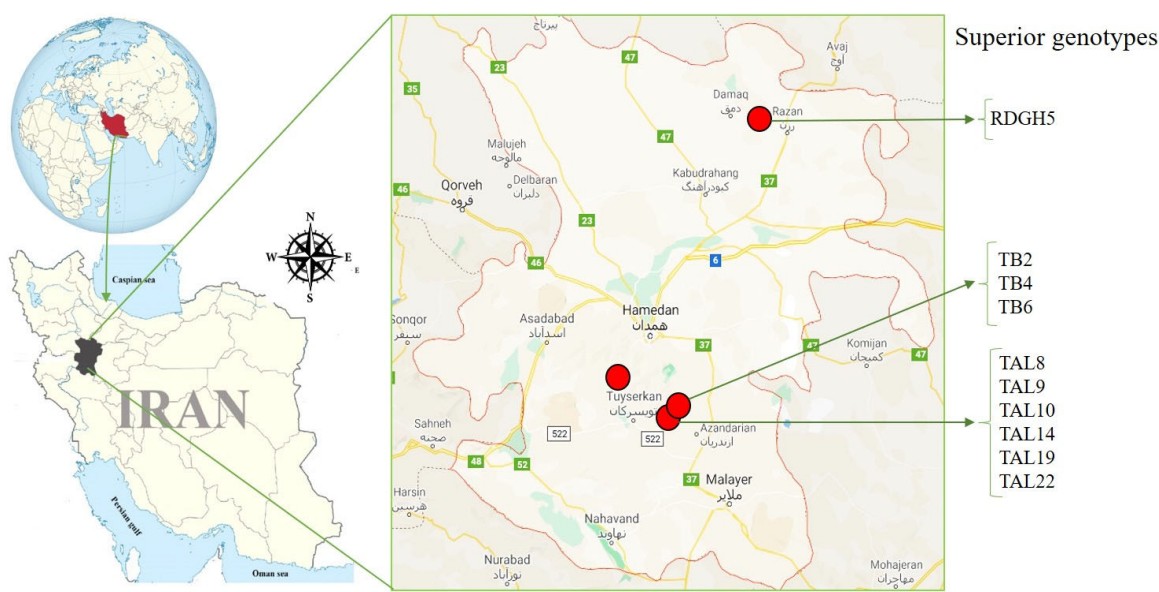

**Figure 1.** Map of walnut germplasm evaluation locales in Hamedan, Iran, and superior genotypes in this research.

**Table 1.** Environmental conditions for the regions of Hamedan Province where walnut genotypes were selected for the study.

| Origin (City) | Latitude | Longitude | Altitude (m) | Annual Rainfall (mm) | High Temp Average (°C) | Low Temp Average (°C) | Annual Temp Average (°C) |
|---|---|---|---|---|---|---|---|
| Tuyserkan | 34°54′98″ | 48°45′37″ | 1780 | 355 | 17.75 | 7 | 12.35 |
| Razan | 35°39′14″ | 49°03′37″ | 1850 | 338 | 16.33 | 5.58 | 10.95 |
| Korzan | 34°54′48″ | 48°35′08″ | 1790 | 361 | 18.08 | 7 | 12.54 |

**Table 2.** Names and origins of the 47 walnut accessions evaluated in this study.

| Genotype | Origin (City) | Genotype | Origin (City) | Genotype | Origin (City) | Genotype | Origin (City) |
|---|---|---|---|---|---|---|---|
| TAL1 | Tuyserkan | TAL19 | Tuyserkan | TB11 | Tuyserkan | KG5 | Korzan |
| TAL2 | Tuyserkan | TAL20 | Tuyserkan | TAL12 | Tuyserkan | KG6 | Korzan |
| TAL3 | Tuyserkan | TAL21 | Tuyserkan | TAL13 | Tuyserkan | KG10 | Korzan |
| TAL4 | Tuyserkan | TAL22 | Tuyserkan | TB2 | Tuyserkan | KG16 | Korzan |
| TAL5 | Tuyserkan | TAL23 | Tuyserkan | TB3 | Tuyserkan | KG18 | Korzan |
| TAL6 | Tuyserkan | TAL24 | Tuyserkan | TB4 | Tuyserkan | KG19 | Korzan |
| TAL8 | Tuyserkan | RD1 | Razan | TB5 | Tuyserkan | KG26 | Korzan |
| TAL9 | Tuyserkan | RD2 | Razan | TB6 | Tuyserkan | KG27 | Korzan |
| TAL10 | Tuyserkan | RD3 | Razan | KG1 | Korzan | KG70 | Korzan |
| TAL11 | Tuyserkan | RD4 | Razan | KG2 | Korzan | KG111 | Korzan |
| TAL14 | Tuyserkan | RDGH5 | Razan | KG3 | Korzan | KG373 | Korzan |
| TAL15 | Tuyserkan | TB1 | Tuyserkan | KG4 | Korzan | | |

### 2.2. Pomological and Phenological Traits

Morphological and pomological characteristics were assessed based on IPGRI (International Plant Genetic Resources Institute) descriptors [6,22,23]. These features included LD and LB, NWT, nut size (i.e., nut length (NL), nut width (NWI), and nut diameter (ND)), shell thickness (STH), packing tissue thickness (PTT), ease of removal of kernel halves (ERKH), shell seal (SS), kernel weight (KW), KP, kernel color (KC), and shape of nut base (SNB) (Figure 2). In order to evaluate leafing date as one of the main traits in this study, the

early-leafing genotype, including KG5, was selected as the reference standard (or 0 day of LD). The LD for this genotype was 15 March. LD was calculated as the average leafing date of genotypes during 2018 and 2019. Twenty nuts were harvested from each genotype and evaluated for two years to generate pomological and trait data.

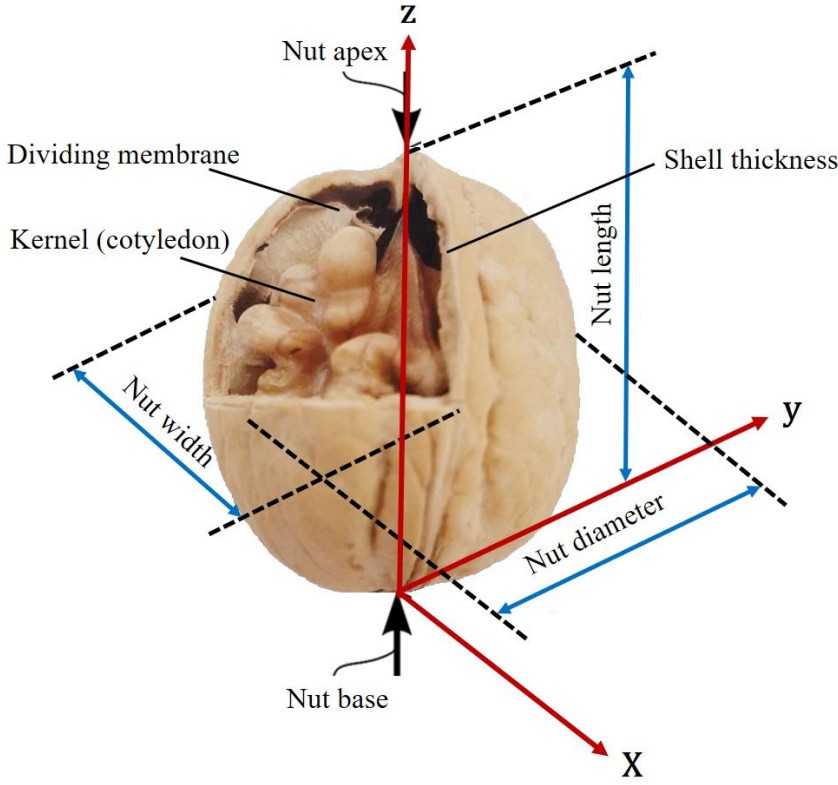

**Figure 2.** Dry fruit of Persian walnut (*Juglans regia*).

### 2.3. Taste Panel

In addition to selecting breeding traits, kernel taste (KT) is also a valuable consideration. Nut taste varies widely among walnut genotypes. A group of five panelists conducted tests that included kernel brittleness (KB) and KT 3 months after harvesting. Panelists were then asked to give a score of 0–10 to any sample tested for the above properties [24,25]. Scoring levels included 0–3 non-frangible/bitter, 4–7 medium, and 8–10 frangible/delicious. A score of 10 was given for samples that were tasty and displayed a brittle kernel.

### 2.4. Grafting of Superior Genotypes

After a 2-year primary evaluation (2018–2019), 47 superior genotypes were selected for detailed pomological and phenological assessment. Of these, 10 superior genotypes were selected based on key breeding traits such as NWI (>12.5 g), KW (>6.5 g), KP (>48%), STH, KC, ERKH, late-leafing, LB (>75%), KT, and PB. Cuttings from superior genotypes were prepared in September and grafted onto seed rootstock in the Pakdasht region of Tehran Province under the same growth conditions.

### 2.5. Statistical Analysis

Pomological and phenological data were evaluated for two years. Descriptive statistical analyses were performed using Excel (Version 2019) and SPSS (Version 22) software, with additional interpretive analyses being completed with R (RStudio 2022). Coefficients of variation (CV = SD/mean × 100) were estimated as the indicator of variability. Using factor analysis and Varimax rotation methodology [26], factor separation was performed. Factor coefficients of 0.4 and above were considered significant for each main independent factor. CA and genotype segregation into groups were performed using the Ward

method of minimum variance based on Squared Euclidean distances calculated after the standardization of data [27]. The R package was used to construct the dendrogram, while Spearman's correlation coefficient was employed to evaluate each genotype for correlations between pomological and phenological traits. Principal component analysis (PCA) was performed using SPSS software in tandem with the R package.

## 3. Results and Discussion

### 3.1. Description of Genotype Characteristics

Recently, breeding techniques have been successfully exploited, and knowledge of phenotypic heterogeneity has been applied to crop species to produce novel cultivars. Leafing date (LD), yield, and kernel percentage (KP), for example, have all been employed as selection factors in walnut breeding efforts [5]. The accession with the earliest leafing date was used to define LD in this study. Walnut production in temperate zones is often hampered by the advent of cold temperatures in late spring. As a result, delay in bud break increases the chances of crop and canopy survival in harsh years [28,29].

Since there was no noticeable difference between the parameters measured in the two years of the experiment, the average data were used. Individual variability was observed for all quantitatively examined traits (Table 3). Therefore, genotypes exhibiting desirable values of a trait could be chosen. PTT had the largest coefficient of variation (CV = 49.93%), whereas LD had the lowest CV (3.22%). The CV values for NWT, kernel weight (KW), and KP were 11.05, 12.64, and 7.30%, respectively. Arzani et al. [30] and Khadivi et al. [31] reported the highest and lowest CVs for STH (29 and 31%, respectively) and NWI (10 and 8%, respectively). NWT ranged from 7.15 to 21.05 g among the individual studied walnuts (Table 3). These results were consistent with those of Shamlu et al. [2]. Sarikhani et al. [7] reported nut weights of 13.2 to 18.8 g, while Mahmoodi et al. [19] reported weights of 6.36 to 15.33 g in similar studies. Diverse average nut weights were described in other investigations as well. For example, Hassani et al. [17] noted an average NWT of 13.62 g, while Akca and Ozongun [1] reported 15.20 g from Anatolia (Turkey).

**Table 3.** Descriptive analysis of examined traits in 47 Persian walnut genotypes. (two-year mean values).

| Trait | Abbr. | Unit | Min. | Max. | Mean | Std. Deviation | Variance | CV% |
|---|---|---|---|---|---|---|---|---|
| Nut weight | NWT | g | 7.15 | 21.05 | 13.01 | 2.54 | 6.48 | 11.05 |
| Nut length | NL | mm | 32.59 | 51.13 | 38.58 | 3.74 | 14.03 | 15.35 |
| Nut width | NWI | mm | 24.79 | 41.20 | 33.98 | 2.48 | 6.17 | 13.66 |
| Nut diameter | ND | mm | 25.00 | 41.73 | 34.51 | 2.98 | 8.93 | 14.27 |
| Kernel weight | KW | g | 3.40 | 10.84 | 6.44 | 1.33 | 1.77 | 12.64 |
| Kernel percentage | KP | % | 30.94 | 69.75 | 49.69 | 5.23 | 27.40 | 7.30 |
| Shell thickness | STH | mm | 0.80 | 2.20 | 1.29 | 0.26 | 0.07 | 12.97 |
| Packing tissue thickness | PTT | mm | 0.04 | 0.80 | 0.19 | 0.12 | 0.01 | 49.93 |
| Shape of nut base | SNB | Code | 3.00 | 9.00 | 7.56 | 1.58 | 2.50 | 11.17 |
| Ease in removal of kernel halves | ERKH | Code | 1.00 | 6.00 | 2.66 | 1.54 | 2.39 | 17.51 |
| Lateral Bearing | LB | % | 50 | 100 | 82.18 | 12.32 | 151.9 | 4.63 |
| Leafing date | LD | Day | 0 | 40 | 25.44 | 9.12 | 83.20 | 3.22 |
| Kernel Brittleness | KB | Code | 4 | 10 | 8.81 | 1.36 | 1.87 | 5.69 |
| Kernel Taste | KT | Code | 4 | 10 | 8.42 | 1.60 | 2.56 | 3.34 |

Walnut KW and KP ranged from 3.40 to 10.84 g and 30.94 to 69.75%, respectively (Table 3). Nut size and NWT may affect KP considerably, as indicated by the strong correlations reported [9,31]. Like NWT, KP is also an indicator of a walnut economic performance in walnut trees [32]. The highest KP seen here (69.75%) was higher than the 63.80% reported by Zeneli et al. [33]. A high KP (>46%) is a primary walnut breeding objective [34,35]. Zeneli et al. [33], Aslantas [29], and Ebrahimi et al. [9] evaluated walnut germplasms in Albania, Turkey, and Iran, respectively. The authors noted the average KP for superior genotypes to be 63.80% for Albania, 67.14% for Turkey, and 62.18% for Iran.

Ideal walnuts should weigh between 12 and 18 g [36] with an optimal KW of 6.00 to 10.00 g, or at least 50.00% of total NWT [31,34]. In walnut breeding programs, a highly desirable KP should be greater than 50.00% [34].

The key objective of this study was to detect late-leafing genotypes with desirable fruit characteristics, and superior trees were selected from all evaluated genotypes. Genotypes with a LD 25 d after the reference standard (early-leafing genotype with March 15 LD), LB > 75%, NWT > 12.5 g, KW > 6.5 g, KP > 48%, STH < 1.2 mm, packing tissue thicknesses (PTT) < 0.22 mm, a light to extra light KC, ERKH, and KT were selected (Table 4).

**Table 4.** Average pomological traits (±SE) from 10 superior walnut genotypes in 2018 and 2019.

| Genotype | NWT (g) | KW (g) | KP (%) | NL (mm) | NWI (mm) | ND (mm) | STH (mm) | PTT (mm) |
|---|---|---|---|---|---|---|---|---|
| TAL8 | 12.76 ± 0.82 | 6.53 ± 0.41 | 51.18 ± 1.44 | 41.42 ± 0.79 | 34.17 ± 0.58 | 33.87 ± 0.46 | 1.08 ± 0.21 | 0.21 ± 0.07 |
| TAL9 | 14.47 ± 2.34 | 7.79 ± 0.97 | 53.83 ± 2.29 | 43.90 ± 0.69 | 35.06 ± 1.32 | 37.94 ± 1.01 | 1.20 ± 0.05 | 0.12 ± 0.02 |
| TAL10 | 14.85 ± 1.33 | 7.84 ± 0.87 | 52.80 ± 1.89 | 38.58 ± 1.09 | 34.71 ± 1.45 | 35.20 ± 1.35 | 1.10 ± 0.9 | 0.24 ± 0.19 |
| TAL14 | 16.82 ± 1.29 | 8.15 ± 1.40 | 48.45 ± 5.36 | 37.58 ± 2.92 | 38.77 ± 1.40 | 40.19 ± 1.09 | 1.13 ± 0.24 | 0.18 ± 0.09 |
| TAL19 | 15.72 ± 1.38 | 8.07 ± 0.74 | 51.37 ± 1.62 | 37.32 ± 1.23 | 34.76 ± 0.75 | 36.44 ± 1.08 | 1.23 ± 0.03 | 0.14 ± 0.02 |
| TAL22 | 15.84 ± 0.56 | 7.98 ± 0.51 | 50.39 ± 1.45 | 38.02 ± 0.46 | 36.60 ± 0.38 | 38.36 ± 0.95 | 1.19 ± 0.08 | 0.11 ± 0.03 |
| TB2 | 12.77 ± 1.00 | 6.87 ± 0.44 | 53.83 ± 1.89 | 40.28 ± 3.18 | 34.23 ± 0.87 | 33.38 ± 0.67 | 1.06 ± 0.12 | 0.13 ± 0.02 |
| TB4 | 13.70 ± 1.52 | 7.56 ± 1.09 | 55.20 ± 2.82 | 46.74 ± 1.88 | 33.48 ± 1.17 | 32.80 ± 1.44 | 1.13 ± 0.06 | 0.14 ± 0.05 |
| TB6 | 12.52 ± 1.30 | 6.79 ± 0.70 | 54.24 ± 1.27 | 37.71 ± 2.12 | 35.06 ± 1.01 | 33.49 ± 1.35 | 1.14 ± 0.21 | 0.09 ± 0.01 |
| RDGH5 | 12.91 ± 0.30 | 6.55 ± 0.69 | 50.71 ± 4.86 | 39.91 ± 1.72 | 31.82 ± 0.96 | 32.63 ± 0.71 | 1.25 ± 0.13 | 0.10 ± 0.03 |
| Genotype | SNB | ERKH | LB | LD | KB | KT | SS | KC |
| TAL8 | 9 | 2 | 100 | 27 (11–April) | 10 | 10 | Triangular | Extra light |
| TAL9 | 7 | 3 | 80 | 28 (12–April) | 9 | 8 | Broad elliptic | Extra light |
| TAL10 | 9 | 1 | 100 | 35 (19–April) | 10 | 10 | Round | Extra light |
| TAL14 | 7 | 3 | 80 | 25 (9–April) | 8 | 9 | Short trapezoid | Light |
| TAL19 | 9 | 1 | 80 | 25 (9–April) | 10 | 10 | Round | Extra light |
| TAL22 | 8 | 2 | 80 | 35 (19–April) | 10 | 10 | Round | Light |
| TB2 | 9 | 1 | 75 | 34 (18–April) | 10 | 10 | Elliptic | Extra light |
| TB4 | 9 | 1 | 90 | 40 (24–April) | 10 | 9 | Elliptic | Light |
| TB6 | 8 | 1 | 85 | 36 (20–April) | 10 | 10 | Short trapezoid | Extra light |
| RDGH5 | 9 | 1 | 80 | 30 (14–April) | 9 | 10 | Broad elliptic | Light |

NW, nut weight; KW, kernel weight; KP, kernel percentage; NL, nut length; ND, nut diameter; STH, shell thickness; PTT, packing tissue thickness; SNB, shape of nut base; ERKH, ease in removal of kernel halves (1 = very easy; 5 = moderate; 9 = very difficult); KB, Kernel brittleness (1 = non-frangible, 10 = frangible); KT, kernel taste (1 = bitter, 10 = delicious); SS, shell shape; KC, kernel color.

One of the most important goals for new cultivars is to improve phenological features, particularly LD. Late-leafing genotypes are not only resistant to late-spring frost but can also grow in mountainous regions where late-spring frosts are common [5]. The superior genotypes selected in this study were mid-leafing and late-leafing. Ten genotypes, TAL8, TAL9, TAL 10, TAL14, TAL19, TAL22, TB2, TB4, TB6, and RDGH5, were considered promising and valuable to future breeding programs. In addition to being late-leafing, these genotypes also exhibited superior pomological characteristics (Table 4).

LD and LB had the highest variation in this study (Table 3). LD was established using the early-leafing genotype in each region and basing all study genotypes on this variable. This study showed superior genotype leafing times to be 25 to 40 d following the reference standard (Table 4). Although the phenological traits (e.g., LD) depend on environmental conditions [37], they are highly heritable [38,39]. Hence, evaluating phenological and other traits with high heritability can be vital to the success of the germplasm evaluation method for the identification of superior walnuts.

STH (0.81–1.25 mm), PTT (0.05–0.13 mm), ERKH, and KC (light-extra light) of the chosen progeny were all within the desirable range for designated superior genotypes with favorable pomological characteristics. These results are consistent with those previously obtained to determine superior walnut genotypes [19,20,31]. Considerable variation in nut size and shape was observed across genotypes (Figure 3).

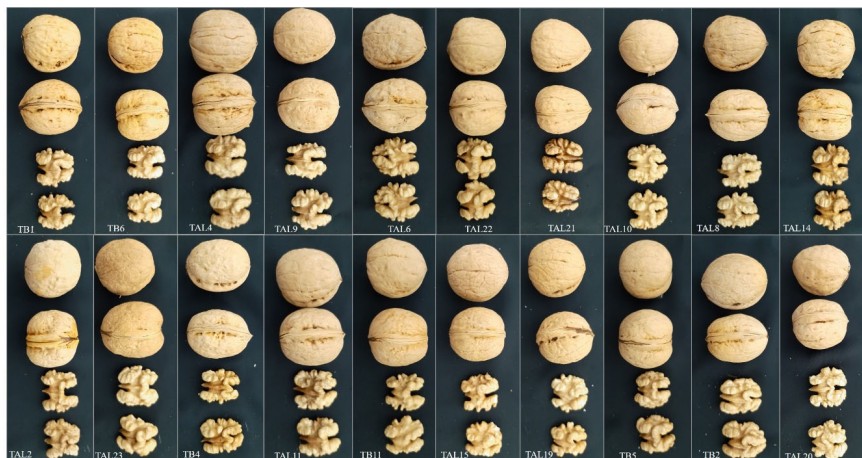

**Figure 3.** Kernels and nuts corresponding to some walnut genotypes in this study were obtained from germplasm evaluation.

### 3.2. Correlations between Traits

Fruit breeders are interested in characterizing pomological traits in walnut breeding programs that are reflective of high heritability. Understanding the strong relationships between pomological traits and other characteristics may help guide walnut breeding efforts [40]. Pomological, phenological, and taste data from different genotypes were used for correlation analysis based on these beliefs. Spearman correlation analysis indicated a strong and positive correlation ($p < 0.01$) between NWT, NWI, ND, and KW (Figure 4). This finding coincides with results from Amiri et al. [41], Sharma and Sharma [40], Sarikhani Khorami et al. [42], and Poggetti et al. [43]. A slightly positive correlation ($p < 0.05$) was observed for NL, LD, and PB. These results mirror those reported by Sarikhani Khorami et al. [42]. A slight positive correlation ($p < 0.05$) was also observed between KT and KW with KP. The strong negative correlation ($p < 0.01$) noted here between STH and PTT with KP is in line with the reports of other researchers [6,19]. An additional negative correlation was observed between ERKH and LD, and KT (Figure 4).

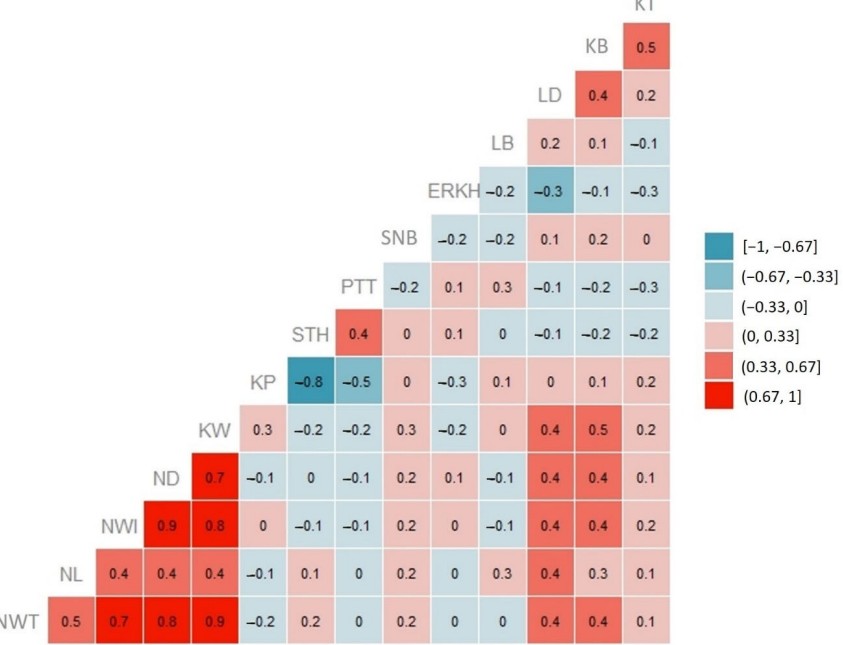

**Figure 4.** Heatmap of Spearman's correlation coefficients between phenological, pomological, and taste traits within the studied walnut genotypes.

### 3.3. Multivariate Analysis

Principal Components Analysis (PCA) is a multivariate statistical method widely used for preprocessing and reducing data size, and the resulting components are used for multivariate statistical analysis. The 14 traits considered in this study were divided into 5 groups or factors using factor analysis and Varimax rotation methodology [26]. Factor analysis revealed that five factors accounted for 76.83% of the total variance (Table 5). Influenced by NWT, NWI, ND, and KW, the first main component could explain 32.10% of the variance; a result consistent with Mahmoudi et al. [19]. The second component, consisting of STH, PTT, and ERKH, accounted for 17.07% of the total variance (Table 5). LB, LD, and PTT made up the third principal component and explained 10.81% of the variance. The fourth and fifth factors each explained less than 10% of the variance. The fourth factor, at 8.68%, was related to KT, STH, and LB, while the fifth factor, at 7.54%, was characterized by KT and LB alone.

**Table 5.** Loadings for the first five phenological trait factors from walnut genotypes and cultivars.

| Trait | Factor | | | | |
|---|---|---|---|---|---|
| | 1 | 2 | 3 | 4 | 5 |
| Nut weight (NWT) | 0.842 | 0.398 | −0.043 | 0.021 | 0.034 |
| Nut length (NL) | 0.53 | 0.245 | 0.308 | −0.198 | −0.164 |
| Nut width (NWI) | 0.828 | 0.2 | −0.206 | −0.080 | 0.054 |
| Nut diameter (ND) | 0.792 | 0.281 | −0.231 | −0.118 | 0.117 |
| Kernel weight (KW) | 0.894 | −0.019 | −0.120 | −0.153 | −0.021 |
| Kernel percentage (KP) | 0.152 | −0.813 | −0.166 | −0.334 | −0.103 |
| Shell thickness (STT) | −0.178 | 0.733 | 0.196 | 0.4 | −0.056 |
| Packing tissue thickness (PTT) | −0.242 | 0.488 | 0.421 | −0.085 | 0.221 |
| Shape of nut base (SNB) | 0.318 | −0.029 | 0.102 | 0.315 | −0.768 |
| Ease of removal of kernel halves (ERKH) | −0.149 | 0.481 | −0.526 | −0.282 | 0.18 |
| Lateral Bearing (LB) | 0.043 | −0.126 | 0.722 | −0.395 | 0.232 |
| Leafing date (LD) | 0.583 | −0.128 | 0.485 | −0.028 | −0.059 |
| Kernel brittleness (KB) | 0.595 | −0.313 | 0.136 | 0.302 | 0.27 |
| Kernel Taste (KT) | 0.302 | −0.385 | −0.033 | 0.653 | 0.443 |
| Variability (%) | 32.1 | 17.07 | 10.81 | 8.68 | 7.54 |
| Cumulative (%) | 32.1 | 49.8 | 60.61 | 69.29 | 76.83 |

Results from other researchers have also shown that a great majority of the traits within the first and second factor groups contributed to higher percentages of variance [7,19]. In this study, the first two primary components accounted for 45% of the initial data variation. Consistent with results from Pope et al. [44] and Rasouli et al. [45], nut and kernel traits were important first and second component factors.

The detection of phenotypic variation among genotypes and accessions, the quantification of morphological and pomological trait correlations, and investigations of various plant genetic resources have been aided by PCA [46]. This study illustrated correlations among select walnut genotypes with PCA. Genotype projections on the PC1 (Dim1)/PC2 (Dim2) plane were based on regression factor scores (Figure 5). The PCA 2-dimension scatterplot displayed phenotypic variability for the genotypes dispersed across the entire plot, emphasizing that these selected genotypes were highly variable. High levels of phenotypic variability in pomological traits have previously been documented in other walnut collections [36,40,47].

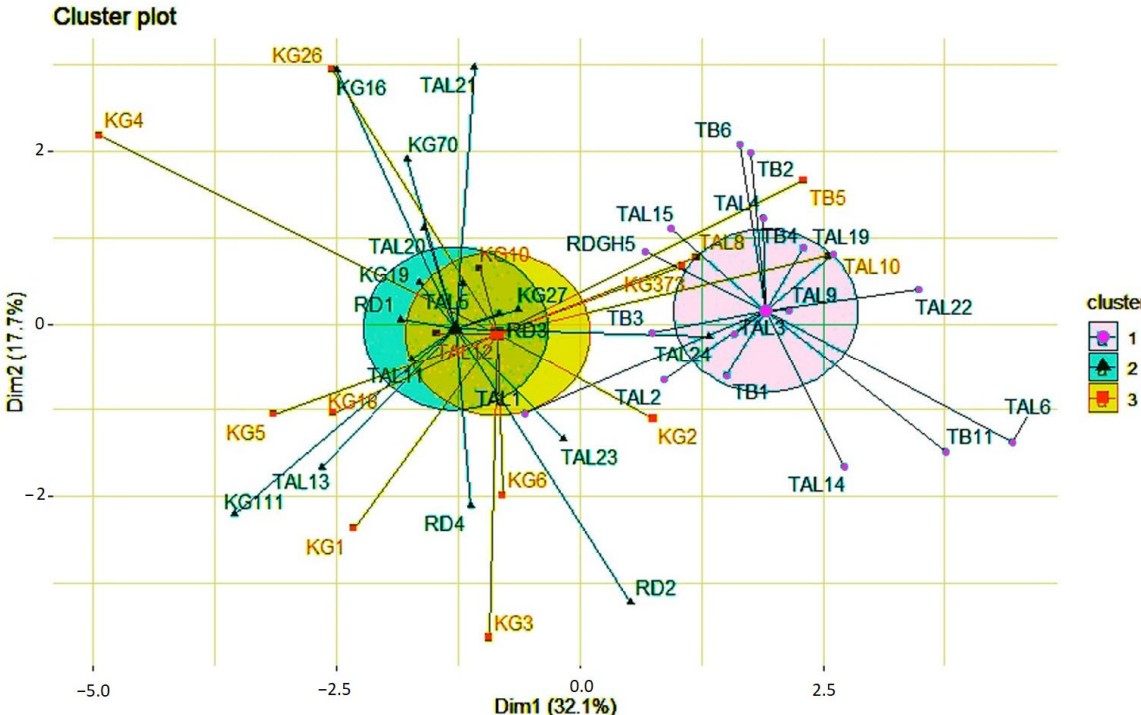

**Figure 5.** Principal Components Analysis (PCA) via distance matrix of 47 walnut genotypes from populations within Hamedan, Iran.

*3.4. Cluster Analysis (CA)*

The optimal number of clusters was calculated using k-means clustering (Figure 6). Using the Ward method and Euclidean distance, cluster analysis detected four major clusters (Figure 7). The first group consisted of 18 genotypes, which mainly had mid-leafing time, very high LB, and mostly small fruit size. The second group consisted of five genotypes with lower STH and PTT. The third group consisted of 18 genotypes with late-leafing time, very high LB, and ideal pomological characteristics. The genotypes in this group usually had the highest KP and lower efficiency in nut production. Most of the superior genotypes selected in this study (i.e., TAL8, TAL 10, TAL19, TAL22, TB2, TB4, TB6, and RDGH5) were present in this group. The fourth group consisted of six genotypes, with the largest fruit size belonging to the genotypes of this group that had a mid to late-leafing time. The superior genotypes TAL9 and TAL14 were in this group.

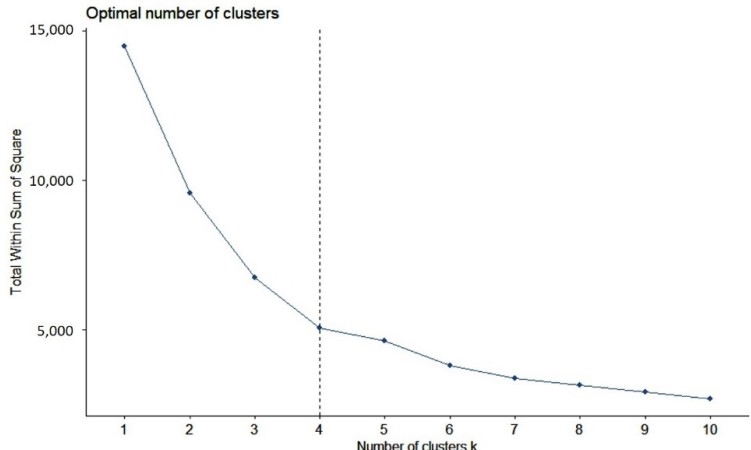

**Figure 6.** Estimated value of k-means for structure analysis and identification of optimal numbers of clusters.

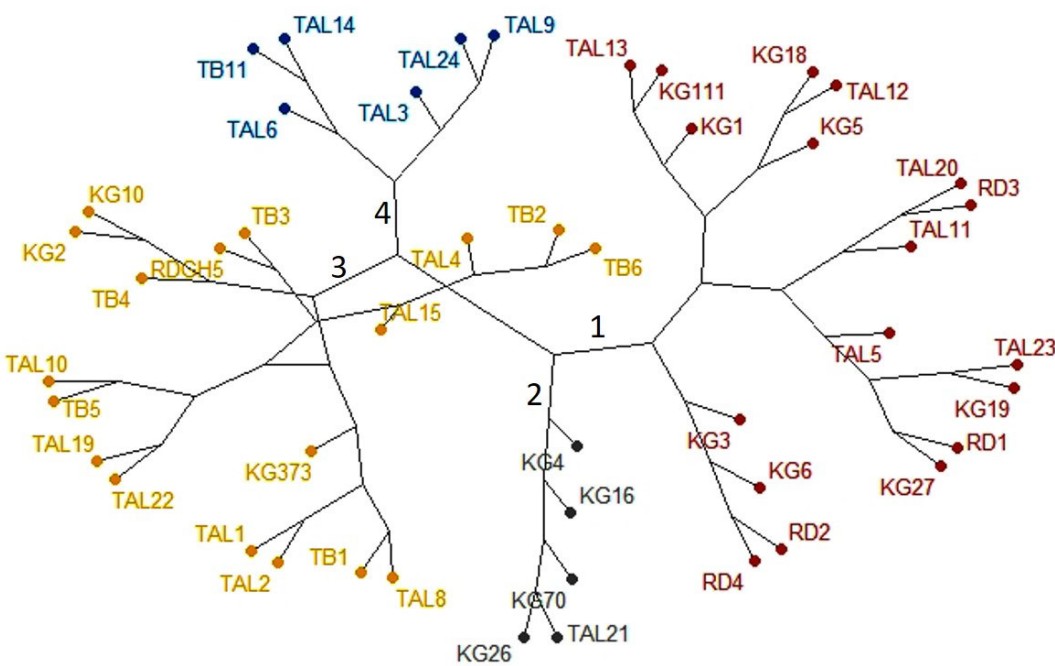

**Figure 7.** Dendrogram illustrating groupings of the Persian walnut genotypes used in this study.

The use of PCA and CA showed some regionally distinct genotypes clustered into the same group. However, walnut populations within the Hamedan Province of Iran revealed substantial genetic diversity after primary phenotypic assessment [31]. Clustering of superior genotypes from different regions into the same group may result from the close geographic distance between the two analyzed regions. Another theory posits the prior use of these genotypes in Hamedan for walnut-orchard development across the province [40].

## 4. Conclusions

The genetic diversity of walnuts in Iran is extensive. Evaluating walnut germplasm from different regions, collecting superior genotypes, and including them in breeding programs can effectively generate new cultivars with desirable phenological and pomological traits. Walnut populations in the Hamedan Province, a major walnut production region in Iran, were studied to isolate superior and late-leafing genotypes characterized by desirable nut and kernel characteristics. High genetic variation was observed in Hamedan's walnut population in terms of different phenotypic characteristics. As a result, TAL8, TAL9, TAL 10, TAL14, TAL19, TAL22, TB2, TB4, TB6, and RDGH5 were introduced as superior genotypes with late-leafing and superior pomological qualities. These newly categorized superior genotypes have been collected and vegetatively propagated for further evaluation. We hope, by continuing this breeding program, to continue efforts to identify additional late-leafing cultivars with desirable nut traits.

**Supplementary Materials:** The following supporting information can be downloaded at: https://www.mdpi.com/article/10.3390/horticulturae8111003/s1, Figure S1a: Diagram of Maximum, Minimum, and Average temperature of Tuyserkan in 2018 and 2019; Figure S1b: Diagram of Maximum, Minimum, and Average temperature of Razan in 2018 and 2019; Figure S1c: Diagram of Maximum, Minimum, and Average temperature of Kurzan in 2018 and 2019.

**Author Contributions:** Conceptualization: K.V., S.S., D.H. and M.R.; Data curation: M.F., D.H., M.R. and S.S.; Formal analysis: M.F. and S.S.; Funding acquisition: K.V. and S.S.L.; Methodology: M.F. and S.S.; Project administration: K.V. and S.S.; Resources: M.R., D.H., S.S.L. and K.V.; Validation: M.R., D.H., S.S.L., S.S. and K.V.; Writing—original draft preparation: M.F.; Writing—review and editing: M.R., D.H., S.S.L., S.S. and K.V. All authors have read and agreed to the published version of the manuscript.

**Funding:** The project is financially supported by the University of Tehran.

**Data Availability Statement:** Not applicable.

**Acknowledgments:** We would like to express our gratitude to all of Hamedan's excellent gardeners, particularly Alvandi, Basiri, and Jalili, who assisted us in this research. We also would like to thank the Iran National Science Foundation (INSF), the Center of Excellence for Walnut Improvement and Technology of Iran, and the University of Tehran for their support.

**Conflicts of Interest:** The authors declare no conflict of interest.

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
