# Peer review of "Tracing Superior Late-Leafing Genotypes of Persian Walnut for Managing Late-Spring Frost in Walnut Orchards"

_horticulturae, doi:10.3390/horticulturae8111003_

Round 1
Reviewer 1 Report
This manuscript presents a study on evaluating genetic diversity in walnut populations. More importantly, superior genotypes with late-leafing and desirable nut traits were screened from initial selected genotypes. Grossly, this work is meaningful and could provide a reference for further extensively application of superior genotypes.
However, the material used in this work only limited in the Hamedan province of Iran, and no results from other regions was compared, thus the conclusion from this study can’t define this area as “one of the richest, most genetically diverse regions in Iran”. And, the description throughout whole MS. is usually rigmarole and suggest to simplify.
Besides, I still have the following main concerns and suggestions:
Topic: This topic represents small part work and ignores majority of this study, more suitable topic should be considered to match the contents
Abstract: Some statements are vague and easy to be misunderstood. e.g. L14-16 stated unclearly, is it means that the present results resulted from the investigations during 2019 - 2020? And this period is not in accord with that stated in Line 81. Also, “(CA) further categorized genotypes into one of four groups” means ??. The last two sentences is suggested to be deleted, for it is meaningless.
Introduction: Review on late-leafing is not sufficient, large publications presented similar results. Thus, high-level overview could emphasize the meaning of this work. It should be pointed out that the primary objective in this study focused on evaluation of superior genotypes rather than on identification of a reliable method.
M & M
1. Pre-evaluations were made on 180 trees among 30-50 year old genotypes from the main walnut-producing areas of Hamedan, natural forest or plantation?
2. L 102, It should be definite which genotype is as the reference standard of early-leafing genotype, for the LD could vary depending on the annual climatic change.
3. No. of nuts used for measuring traits is not enough for assessing the selected genotypes, because of its high variation within individual
4. 2.3 Taste panel, suggested to list the scored levels and corresponding taste description.
R & D
1. The first para. in 3.1 can be simplified and emphasized.
2. L 145 that …genotypes exhibiting different values of a trait could be chosen” can’t provide clear basis for selection.
3. Table 3, check the unit carefully.
4. How to calculate out the 0 day of LD should be expressed in detail in 2.2
Author Response
Dear reviewer #1, Thank you so much for your motivational feedback as well as valuable comments. All changes/additions in the manuscript file are highlighted in yellow.
Topic: This topic represents small part work and ignores majority of this study, more suitable topic should be considered to match the contents
Based on your comment, the title “Tracing superior and late-leafing genotypes in their origin center for managing late-spring frost in walnut orchards” changed to “Tracing superior late-leafing genotypes of Persian walnut for managing late-spring frost in walnut orchards”
Abstract:
Some statements are vague and easy to be misunderstood. e.g. L14-16 stated unclearly, is it means that the present results resulted from the investigations during 2019 - 2020? And this period is not in accord with that stated in Line 81. Also, “(CA) further categorized genotypes into one of four groups” means ??. The last two sentences is suggested to be deleted, for it is meaningless.
The sentence related to line 14-16 was corrected. The mentioned sentence was revised based on your comment. Also, the last two sentences of the abstract section were deleted
Introduction: Review on late-leafing is not sufficient, large publications presented similar results. Thus, high-level overview could emphasize the meaning of this work. It should be pointed out that the primary objective in this study focused on evaluation of superior genotypes rather than on identification of a reliable method.
Some literatures on late-leafing and late-spring frost were added to introduction section. So that, three new references were added and reference numbers were changed throughout the manuscript.
M & M
Pre-evaluations were made on 180 trees among 30-50 year old genotypes from the main walnut-producing areas of Hamedan, natural forest or plantation?
The selected genotypes were seed-originated trees which is planted in traditional orchards. This sentence was added to M & M section:
This research was conducted in the Hamedan province of Iran, a leading center of walnut genetic diversity in 2018 and 2019. 180 seed-originated walnut trees with 30-50-years-old genotypes located in traditional orchards were selected and pre-evaluated. The studied genotypes were selected from the main walnut-producing areas of Hamedan, including Tuyserkan, Razan and Korzan based on phenotypic traits
L 102, It should be definite which genotype is as the reference standard of early-leafing genotype, for the LD could vary depending on the annual climatic change.
‘KG5’ genotype was used as the reference standard which is mentioned in Materials and Methods. LD is based on average data of two years.
No. of nuts used for measuring traits is not enough for assessing the selected genotypes, because of its high variation within individual
Dear reviewer, based on the “section 6.5 of walnut descriptor (IPGRI)”, 20 nuts is needed for evaluating nut related traits. We also used 20 nuts for evaluating nut related traits based on IPGRI descriptors as was addressed in the previous studies. In this study and our previous studies, we considered 5 replications and 4 nuts in each replication. We didn’t see any significant difference between replications in term of nut related traits. Considering that we evaluated the genotypes for two consecutive years, we evaluated 40 nuts for each genotype (20 fruits in each year) to assess the selected genotypes.
2.3 Taste panel, suggested to list the scored levels and corresponding taste description.
Based on your comment, the sentence ‘Scoring levels included 0-3 non-frangible/bitter, 4-7 medium and 8-10 frangible/ delicious’ was added to the taste panel (2.3) section
R & D
The first para. in 3.1 can be simplified and emphasized.
Done
L 145 that …genotypes exhibiting different values of a trait could be chosen” can’t provide clear basis for selection.
Done
Table 3, check the unit carefully.
Based on your comment, the unit related to Nut weight (g) and Kernel percentage (%) was revised.
How to calculate out the 0 day of LD should be expressed in detail in 2.2
LD was calculated as the average leafing date of genotypes during 2018 and 2019. This sentence was added to section 2.2

Reviewer 2 Report
The authors evaluated pomological and phenological traits of different walnut genotypes from the principal Juglans regia genetic diversity center.
The main objectives were to identify genotypes with superior fruit characteristics, and a late bud burst to be less susceptible to late frosts.
The title and abstract are appropriate for the content of the text.
The introduction section provides useful information to the readers and adequately explains the context of the research.
The methods are satisfactory, adequate, and validated.
The paper is of interest for breeding and genetic resources characterization.
My only comment is that it is not specified whether there were any variations in the measured parameters between the two years of testing.
Please include at least one sentence in the results stating that there was no noticeable vintage effect.
Author Response
Reviewer #2:
My only comment is that it is not specified whether there were any variations in the measured parameters between the two years of testing.
Please include at least one sentence in the results stating that there was no noticeable vintage effect.
Dear reviewer, we would like to thank you very much for your precise and valuable comment. Based on your comment and our obtained results, the sentence ‘Since, there was no noticeable difference between the parameters measured in the two years of the experiment, the average data were used (Tables 3 and 4)’ was added to the beginning of the second paragraph of the results and discussion.
